# Hydrogen bond symmetrisation in D$_2$O ice observed by neutron diffraction

Kazuki Komatsu [1] ✉, Takanori Hattori [2], Stefan Klotz [3] ✉,
Shinichi Machida [4], Keishiro Yamashita [1,6], Hayate Ito[1], Hiroki Kobayashi [1],
Tetsuo Irifune [5], Toru Shinmei [5], Asami Sano-Furukawa [2] & Hiroyuki Kagi [1]

Hydrogen bond symmetrisation is the phenomenon where a hydrogen atom is located at the centre of a hydrogen bond. Theoretical studies predict that hydrogen bonds in ice VII eventually undergo symmetrisation upon increasing pressure, involving nuclear quantum effect with significant isotope effect and drastic changes in the elastic properties through several intermediate states with varying hydrogen distribution. Despite numerous experimental studies conducted, the location of hydrogen and hence the transition pressures reported up to date remain inconsistent. Here we report the atomic distribution of deuterium in D$_2$O ice using neutron diffraction above 100 GPa and observe the transition from a bimodal to a unimodal distribution of deuterium at around 80 GPa. At the transition pressure, a significant narrowing of the peak widths of 110 is also observed, attributed to the structural relaxation by the change of elastic properties.

Water ice exhibits remarkable structural variety with at least 20 polymorphs discovered up to date[1]. But at pressures above 2 GPa, the structural diversity of ice polymorphs appears to be diminished: the phase diagram of ice is dominated by body-centred cubic ices with hydrogen-disordered ice VII, ordered ice VIII and hydrogen-bond (H-bond) symmetrised ice X, and recently discussed superionic ices having body-centred cubic and face-centred cubic structures. The concept of H-bond symmetrisation in ice, *i.e.*, hydrogens located at the centre between donor and acceptor oxygens in the H-bond, was already pointed out more than half a century ago by Kamb and Davis in the first paper on the structure of ice VII published in 1964 as existing 'well above 22 GPa'[2]. In 1996 and 1997, the H-bond symmetrisation in ice was observed experimentally by infrared or Raman spectroscopy at pressures of around 60 GPa for H$_2$O and 70 GPa for D$_2$O[3–5]. The significant isotope effect on the phase boundaries from ice VII to ice X can be explained by the quantum tunnelling of hydrogen (or deuterium) through the energy barrier between the two potential minima, and the nuclear quantum tunnelling effect should be more prominent in

hydrogen than in deuterium[6]. Although the above-quoted transition pressures had long been accepted, recently reported values derived from [1]H-NMR[7] or x-ray diffraction[8,9] diverge significantly from 20 –75 GPa.

A part of the controversy in the previously reported H-bond symmetrisation pressures could be due to various intermediate states between ice VII and X within the same space group $Pn\bar{3}m$, and the fact that none of the experimental techniques applied so far could verify the position of hydrogen in the ice lattice at pressures in the megabar range. At low pressures with relatively long O...O distances (hereafter, $d$(O...O)), hydrogen localises to one side of two potential minima and forms a covalent bond with the closer oxygen atom and weakly interacts with the acceptor oxygen. However, hydrogens in ice VII are spatially disordered in two potential minima which correspond to two crystallographically equivalent sites in H-bonds. They are observed as bimodal distribution in the atomic distribution map obtained from diffraction methods as time- and space-averaged structure as shown in Fig. 1. With increasing pressure, $d$(O...O) decreases and hydrogens

[1]Geochemical Research Center, Graduate School of Science, The University of Tokyo, 7-3-1 Hongo, Bunkyo-ku, Tokyo 113-0033, Japan. [2]J-PARC Center, Japan Atomic Energy Agency, 2-4 Shirakata, Tokai, Naka, Ibaraki 319-1195, Japan. [3]Institut de Minéralogie, de Physique des Matériaux et de Cosmochimie, CNRS UMR 7590, Sorbonne Université, F-75252 Paris, France. [4]Neutron Science and Technology Center, CROSS, 162-1 Shirakata, Tokai, Naka, Ibaraki 319-1106, Japan. [5]Geodynamics Research Center, Ehime University, 2-5 Bunkyo-cho, Matsuyama, Ehime 790-8577, Japan. [6]Present address: Institute of Physical Chemistry, University of Innsbruck, A-6020 Innsbruck, Austria. ✉e-mail: kom@eqchem.s.u-tokyo.ac.jp; Stefan.Klotz@upmc.fr

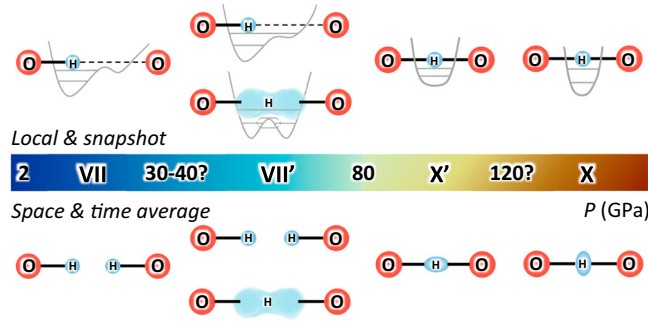

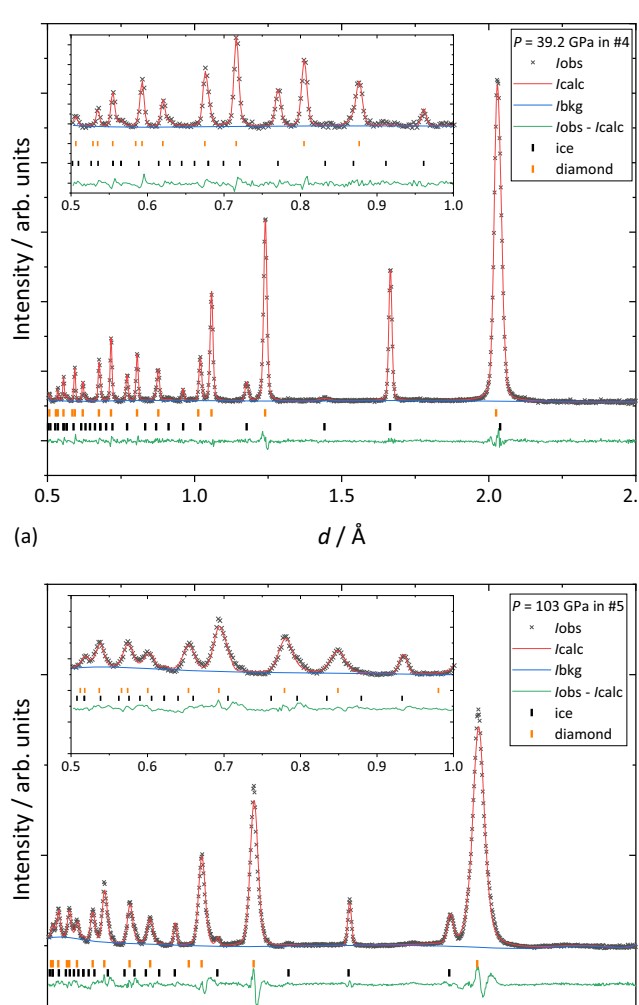

**Fig. 1 | Schematic illustrations for the structural evolution from D₂O ice VII to ice X at room temperature, including intermediate states ice VII' and X'.** The coloured bar represents the pressure scale with approximate transition pressures. Grey curves in local and snapshot illustrations (above the coloured bar) represent potential surfaces and grey horizontal lines are ground and 1st excited energy levels. Two illustrations for ice VII' represent dynamically disordered states by thermal hopping (upper) and quantum tunnelling (lower). Note that the transition pressure from ice VII' to X' are determined to be around 80 GPa by structural analyses of neutron diffraction investigated in this study (see details in text), whereas the other transition pressures are not fully clarified. The VII-VII' transition pressure is derived from a previous $^1$H-NMR study for H₂O ice considering the isotope effect, and the X'-X transition pressure is estimated from a crossing point of the extrapolation lines of $d$(O-D) and $d$(D...O) (see details in text).

start to move to the adjacent minimum by thermal hopping or quantum tunnelling, *i.e.* the H-disorder becomes more dynamic. This state is commonly referred to as ice VII'[10,11]. It is hard to distinguish between static and dynamic disorder in ice VII/VII' based on diffraction methods and also to determine whether the source of hydrogen/deuterium delocalisation in ice VII' is thermal hopping or quantum tunnelling since intensities of Bragg reflections reflect the time- and space-averaged structure. The term 'H-bond symmetrisation' refers to hydrogen/deuterium localised at the centre of two neighbouring oxygens. At relatively low pressure, when the potential shape just starts to change from a double to a single minimum, the hydrogen/deuterium distribution would still elongate along the O...O line, sometimes referred to as ice X'[9]. From first-principles molecular dynamics simulations, it is expected that the ellipsoidal hydrogen distribution would change from parallel (prolate distribution, ice X') to perpendicular (oblate distribution, ice X) to the O...O line at higher pressure[12].

Neutron diffraction would be the most direct method to observe the atomic distribution, and could "in principle" distinguish ice VII' and ice X (X'), but neutron diffraction experiments under pressure above 30 GPa have long been limited mainly due to the weakness of neutron sources[1]. Owing to strong spallation neutron sources constructed in the last decades and the technical developments of high-pressure cells and neutron optics/shielding, the achievable pressure for neutron diffraction have been extended up to 100 GPa very recently[13]. We have also developed diamond anvil cells using nano-polycrystalline diamonds (NPDACs) as the NPDs have promising properties in terms of both hardness and toughness owing to the absence of cleavage. The polycrystalline feature is also beneficial for obtaining accurate intensities as relatively simple attenuation correction can be applied compared to the case using single crystal diamonds[14].

Here we report the accurate atomic distribution of deuterium in D₂O ice above 100 GPa by neutron diffraction, and discuss the long-standing problem of the H-bond symmetrisation.

## Results and discussion

Seven independent runs (#1–#7) of neutron diffraction experiments for D₂O ice at room temperature with varying pressure are reported here. Fine powder samples good enough for the Rietveld analyses were

**Fig. 2 | Results of Rietveld refinements of neutron diffraction patterns for D₂O ice. a** at 39.2 GPa in run #4 and (**b**) at 103 GPa in run #5. $I$obs, $I$calc and $I$bkg means observed, calculated and background intensities, respectively.

obtained in runs #4–#7 by passing through several phase transitions at low temperature (see details in Methods section). The best quality of neutron diffraction patterns was obtained up to 39.2 GPa in run #4, whereas the highest pressure of 106 GPa was achieved in run #5 as shown in Fig. 2. The full structure analyses by the Rietveld methods were performed in runs #4 and #7 where atomic coordinates of deuterium $x$(D) at site 8e, ($x$, $x$, $x$) and isotropic atomic displacement parameters (ADPs) for deuterium and oxygen, $U$(D) and $U$(O), respectively, were refined as variable parameters. The oxygen atom was fixed at a single site 2a $\left(\frac{1}{4}, \frac{1}{4}, \frac{1}{4}\right)$. Although multi-site disorder models for the oxygen were proposed[15,16], the displacement of oxygen from the 2a site was too small to elucidate from the limited resolution data. Thus, the multi-site disorder of oxygen was ignored in this study, and in fact, it is known from our recent neutron diffraction study[17] that the single-site model well reproduces the observed diffraction pattern at 298 K, 2.2 GPa. In other runs #5 and #6, only $x$(D) was refined as a variable parameter with ADPs fixed to values estimated by extrapolating the pressure dependence of ADPs of runs #4 and #7 (Supplementary Fig. 9) by an empirical exponential function. This was done since the diffraction patterns in runs #5 and #6 were of poorer quality than those in #4 and #7 due to differences in sample volume (see Supplementary Table 1), and the small deviation of $U$(D) does not

contribute sufficiently to the diffraction intensities to allow stable refinements. Full details of methods and experimental conditions for each run are described in the method section and Supplementary Table 1, respectively.

The absolute values of ADPs obtained from the Rietveld analyses for run #4 are well consistent with the previously reported values[16–18], in particular with the values reported in Yamashita et al.[17] for ice VII at 2.2 GPa and 298 K (Supplementary Fig. 9). Their powder neutron diffraction data should be the most accurate among all previously reported data for ice VII, derived from 33 independent and 24 duplicated reflections without significant contamination from parasitic scattering from high-pressure cell components. The consistency of our data to their values for ADPs supports the accuracy of diffraction intensities obtained in this study.

At higher pressures, the refined ADPs for deuterium and oxygen in run #4 decreased with increasing pressure up to 15 GPa and shows essentially a plateau above, as previously observed in Nelmes et al.[16] (Supplementary Fig. 9). The observed trend for ADPs could be related to the recently reported characteristic atomic distribution of deuterium in ice VII by Yamashita et al.[17]. They revealed from in-situ single-crystal and powder neutron diffraction measurements that deuterium has a ring-like distribution, which is attributed to the rotation dynamics of the water molecule in this phase. Note here that ice VII exhibits several anomalies in the 10–15 GPa range in Raman spectra[19], x-ray diffraction[20], electric conductivities[21], x-ray induced dissociation[22], proton diffusion[23], and phase transition rate[24]. Yamane et al.[25] recently confirmed from dielectric measurements that these anomalies originate from a crossover of proton dynamics from molecular rotation to proton translation. The larger $U(D)$ at lower pressure and the following decrease up to 15 GPa observed in Yamane et al.[25] is consistent with the interpretation that the large $U(D)$ is the result of the ring-like distribution, and it decreases with increasing pressure because of the suppression of the molecular rotation dynamics. A similar trend for $U(O)$ would also originate from the suppression of rotation dynamics.

The covalent bond, $d(O\text{-}D)$, and H-bond distances, $d(D...O)$, are derived from the refined atomic coordinate, $x(D)$, and the unit cell parameter $a$ of ice as shown in Fig. 3a. The derived bond distances are consistent with the previous neutron diffraction study up to 62 GPa by Guthrie et al.[26], and also show excellent consistency with very recently reported results from path integral molecular dynamics calculation by Tsuchiya et al.[27]. Although the obtained $d(O\text{-}D)$ and $d(D...O)$ do not merge yet up to the pressure investigated of 106 GPa, this does not mean the atomic distribution of deuterium is still bimodal at this pressure. In fact, the probability density function of the atomic distribution of a single atom $P_1(r)$ can be approximated by,

$$P_1(r) = \frac{1}{(2\pi U)^{\frac{3}{2}}} \exp\left(-\frac{r^2}{2U}\right) \quad (1)$$

where $r$ is the distance from the centre of atomic distribution and $U$ is the isotropic ADP. When there are two symmetrically equivalent sites with half occupancy at a distance, $d$, $e.g.$, positions of two sites at $-d/2$ and $d/2$ along the $x$-axis, the probability density function of the atom projected on the $x$-axis, $P_2(x)$, can be written as following:

$$P_2(x) = \frac{1}{2(2\pi U)^{\frac{1}{2}}} \exp\left(-\frac{(x+d/2)^2}{2U}\right) + \frac{1}{2(2\pi U)^{\frac{1}{2}}} \exp\left(-\frac{(x-d/2)^2}{2U}\right) \quad (2)$$

Whether the atomic distribution is unimodal or bimodal can be judged from the sign of $2^{nd}$ derivative of $P_2(x)$ at $x = 0$, $\frac{d^2 P_2(x)}{dx^2}\big|_{x=0}$, and the sign of $\frac{d^2 P_2(x)}{dx^2}\big|_{x=0}$ can be determined by the comparison between $d$ and $2\sqrt{U}$ (see more details in Supplementary Information 6 for this derivation). When $d > 2\sqrt{U}$ the atomic distribution is bimodal, whereas

$d < 2\sqrt{U}$ means unimodal, so that, based on this criterion, the transition pressure from ice VII' to X' for $D_2O$ ice can be defined as the pressure where $d(D...D) = 2\sqrt{U(D)}$.

The observed $d(D...D)$ and $2\sqrt{U(D)}$ crosses at around 80 GPa as shown in Fig. 3b, which is the first direct observation of the ice VII'-X' transition from neutron diffraction. The probability densities of atomic distribution of deuterium atoms as a function of the distance from oxygen and of pressure are shown as a coloured map in Fig. 3a, and the map also demonstrates the distribution change from bimodal to unimodal occurs at around 80 GPa. The above-defined criterion for the transition from ice VII' to X' is based on the assumption that the atomic distribution of deuterium can be regarded as isotropic, or in other words, the deviation from the spherical form such as the ring-like distribution observed at low-pressure region[17] can be ignored, and also the assumption that the $U(D)$ can be extrapolated to higher pressure from run #7 obtained up to 67.5 GPa. We consider that a minor deviation from the spherical distribution or differences from the extrapolation will not drastically change the interpretation. We note that a spherical atomic distribution of two D-sites at distances $-d/2$ and $d/2$ is, at around 80 GPa and above, indistinguishable by diffraction measurements of a single H atom at the centre, but with an ellipsoidal atomic distribution with its long axis along the O...O direction. We conducted such refinements with varying anisotropic ADP for the data taken at 103 GPa in run #5, yielding $U_{\parallel}(D) = 0.021(2)$ Å$^2$, $U_{\perp}(D) = 0.017(2)$ Å$^2$, showing almost spherical or slightly elongate distribution along the O...O direction. The $\chi^2$ of the refinement based on the anisotropic ADPs is almost identical to that based on the two-site model with isotropic ADP (see more details in Supplementary Information 6).

We also found peak widths exhibiting an anomaly at around 80 GPa which indicates the transition. We plotted peak widths divided by $d$-spacings ($\Delta d/d$) for 110 and 111 in Fig. 3c. The $\Delta d/d$ for both 110 and 111 are initially around 0.007, which is almost identical to the instrument resolution of the diffractometer[28]. Both peaks are broadened with increasing pressure, while the peak broadening is more significant for 110 than for 111 as previously observed in x-ray diffraction[20,29,30]. Somayazulu et al. first indicated the possibility of a lowering of symmetry from cubic to tetragonal at around 14 GPa from high-resolution x-ray diffraction[20]. Grande et al. recently reported x-ray diffraction for once-melted ice VII by laser-heating and they proposed the term "ice-VII$_t$" for ice VII with a tetragonal symmetry found at 5.1(5) GPa[30], much lower pressure than previously proposed[20,29].

Here we note that such peak broadening or even peak splitting associated with apparent tetragonal distortion could be induced by external deviatoric stress (see Supplementary Information for more details). This kind of anisotropic broadening is implemented in TOF profile function 4 in the GSAS software[31], which was applied in the Rietveld refinement in this study and drastically improved the fits, in particular for the data taken at around 20 – 30 GPa as shown in Supplementary Fig. 13. The previously reported peak broadening or splitting in x-ray diffraction could also be explainable by the anisotropic microstrain, and some doubt arises whether the proposed symmetry lowering is an intrinsic behaviour of ice VII or just due to the external deviatoric stress. Note here that the peak widths in run #2 using the flat shape of culet with 1.0 mm diameter are significantly larger than those in run #1 using the cupped shape of culet with the same diameter (Fig. 3c) so that the external deviatoric stress should affect at least partly the anisotropic broadening.

On the other hand, another x-ray diffraction study by Lei et al. showed a clear peak splitting for $H_2O$ ice VII with $H_2$ or He as a pressure-transmitting medium[8]. They interpreted these split peaks with higher and lower $d$-spacing as ice VII and ice X, respectively, with still cubic symmetry. They reported ice VII and X coexist in the pressure range from 20 – 50 GPa in $H_2O$ with $H_2$ medium, and from 60 to 70 GPa with He medium. Our result on the atomic distribution requires a revision of

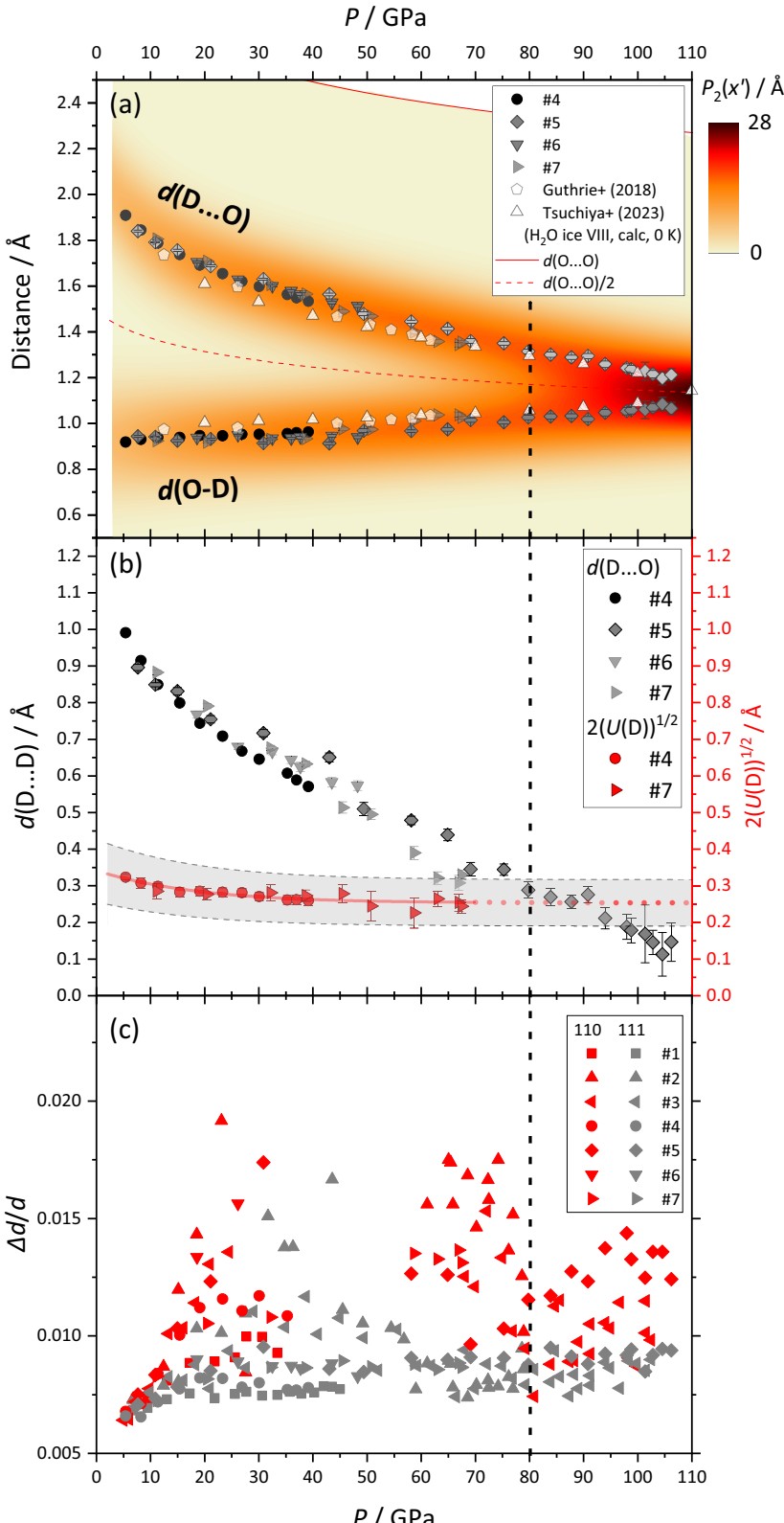

**Fig. 3 | Results of structure analyses for D₂O ice from neutron diffraction up to 106 GPa. a** Atomic distances, $d$(O-D) and $d$(D...O), on a coloured map representing the probability density function for deuterium atom as a function of the distance from oxygen atom, $P_2(x')$, where $x' = x - d(\text{O}...\text{O})/2$ in Eq. (2). The $d$(O...O) and $d$(O...O)/2 are also shown as reference as red line and red dotted line, respectively, which are calculated from the equation of states of ice VII[39]. **b** The distance between two symmetric deuterium sites $d$(D...D) (left axis) compared with two times square root of isotropic ADP for deuterium, $2\sqrt{U(D)}$, for a criterion of the transition from bimodal to unimodal atomic distribution. Red line shows an empirically fitted line for the observed $U$(D) (see Supplementary Fig. 9) converted to $2\sqrt{U(D)}$ with its extrapolation as dotted line and the estimated confident limits (± 25% of the fitted line) as grey shaded area (see discussion in Supplementary Information 2). **c** Peak widths (full widths at half maximum) normalised by $d$-spacing, $\Delta d/d$, plotted as a function of pressure. In the range of 35 − 60 GPa, the 110 reflections could not be observed due to the severe overlap with 111 peaks of diamond (anvil). Data points are represented as refined values with estimated standard deviation as error bars.

their interpretation that the two peaks are indexed to be ice VII and X, since it is highly unlikely that hydrogen atom centreing is achieved at 20 GPa even considering the isotope effect. The difference in the coexisting pressure region of two peaks in $H_2$ and He media implies that the peak splitting is not an intrinsic property of ice VII, but depends on the inhomogeneous stress state in the sample chamber.

The $\Delta d/d$ for both 110 and 111 have a maximum at around 20 GPa (Fig. 3c) as previously reported in the x-ray diffraction study by Somayazulu et al.[20], and this anomaly could be associated with the crossover of proton dynamics mentioned before. Recently, Meier et al.[32] reported that the pressure dependence of the NMR line widths for various H-bond systems including ice VII shows a minimum, associated with a maximum in hydrogen mobility, at a condition that H-bond distance $d(O...O)$ is in a narrow range between 2.44 and 2.45 Å, regardless the chemical environment of the O-H...O unit. Meier et al. also reported in an earlier NMR study that the high hydrogen mobility originated from the quantum tunnelling which could occur above 20 GPa in $H_2O$ ice VII[7]. The structural relaxation, observed as the reduction of $\Delta d/d$, could be linked with the high mobility owing to the quantum tunnelling. Interestingly, another drastic reduction of $\Delta d/d$ is found at around 80 GPa, coinciding with the pressure at which the deuterium distribution changed from bimodal to unimodal (Fig. 3). In the transition from ice VII′ to X′, a slight decrease and subsequent increase in the bulk modulus was recently observed in continuous volume measurements using dynamic diamond anvil cell by synchrotron time-resolved x-ray diffraction[9], and also by Brillouin Scattering[33], whereas a stepwise increase in bulk and shear moduli is expected from ice VIII to X transition from first-principles calculation[34]. The sudden reduction in $\Delta d/d$ observed in this study suggests that a structural relaxation is associated with the change in bulk and shear modulus in the phase transition from ice VII′ to X′.

The extrapolation of the observed $d(O-D)$ and $d(D...D)$ in this study merges at approximately 120 GPa, which should correspond to the hydrogen distribution changing from parallel (prolate) to perpendicular (oblate), i.e., the state transition from ice X′ to X in a narrow definition (see Fig. 1), as initially expected from first-principle molecular dynamics calculation by Benoit and Marx[12]. More recent DFT calculations for $H_2O$ ice by Trybel et al.[35] also showed a gradual change from the double- to a single-well potential, in which symmetrisation starts above 90 GPa accompanying a significant drop of proton jump frequencies, and fully symmetric single well potential is expected above 130 GPa. These "transition" pressures are consistent with our observations with the crossing pressure point of $d(D-D)$ and $2\sqrt{U(D)}$ at around 80 GPa, and the merging point of the extrapolated $d(O-D)$ and $d(D...D)$ at around 120 GPa, respectively. Trybel et al. noted that their calculation did not show any sharp phase transition in the pressure range from 2 GPa to 200 GPa, i.e., at least there is no structural phase transition of first or second order. On the other hand, recent bulk modulus measurements by x-ray diffraction[9] and a combination of density functional theory and molecular dynamics simulations[36] for $H_2O$ ice revealed a distinct change in the pressure dependence of the isothermal bulk modulus at 90–110 GPa, which is also interpreted as the transition from ice X′ to X. The transition pressures from ice X′ to X are again roughly consistent considering the isotope difference since $D_2O$ is expected to have a higher transition pressure than $H_2O$. As indicated by Trybel et al.[35], it is worth pursuing the issue of how sharp structural or thermodynamic parameters changes can be observed in the stepwise changes from ice VII to X including intermediate ice VII′ and X′. The current understanding would be that the VII′-X′ transition may be second order since the elastic constants show discontinuity[27], whereas the VII-VII′ and X′-X transitions could be third order or higher since the change is just related to the pressure dependences of bulk moduli[9].

In conclusion, our neutron diffraction measurements to above 100 GPa for the first time directly observed the H-bond symmetrisation

as centring of the deuterium distribution in ice at around 80 GPa, associated with structural relaxation observed as the sharpening of peak widths. The detailed hydrogen positions with ADPs obtained in this study will contribute to revisiting interpretations of previously reported observations, although several questions have remained, e.g., why x-ray Bragg peaks were split at different pressures in different pressure transmitting media[8], why the tunnelling mode was observed in the NMR measurements up to 97 GPa[7] where the potential curve would have a single-minimum. We finally point out that most of the previous studies have been carried out on $H_2O$ samples, which means that the isotope effect on the successive state changes from ice VII to X has not been investigated. Our comprehensive experiments on $D_2O$ ice, as well as previous measurements on $H_2O$ ice, should contribute to untangling the current inconsistencies.

## Methods

Neutron diffraction experiments were conducted at the PLANET[28] beamline at the Materials and Life Science Experimental Facility (MLF) of J-PARC, Ibaraki, Japan. The incident beam comprised 25 Hz pulsed spallation neutrons with an accelerator power of 500–800 kW depending on experiments. More than 10 independent loadings for the $D_2O$ sample (99.9%, MagniSolv™) into nano-polycrystalline diamond anvil cell (NPDAC) were conducted, and in total 7 runs (#1-#7) had sufficient quality for detailed analyses. In each run, we varied the culet size and shape of anvils, the material and thickness of the gasket, diffraction geometry and neutron collimation using fine radial collimators[37] (see details in Supplementary Table 1). Except for run #2, we used cup-shaped culets to reduce unwanted scattering from the diamond anvils and to reduce deviatoric stress by relatively uniform compression compared to the case using the flat anvils[14,38]. In runs #1-#3, ice VII samples were just prepared by direct compression at room temperature, but we realised that the obtained ice was coarse-grained and not suitable for Rietveld analyses. Thus, in runs #4-#7, we prepared samples through low temperatures, i.e., (i) $D_2O$ water was loaded in the NPDAC and clamped nearly at ambient pressure, (ii) the whole NPDAC was cooled down by dry ice or liquid nitrogen, (iii) compressed at low temperature, and (iv) reheated to room temperature under pressure. This procedure passes through several phase boundaries of ice phases and allows us to make powder samples of ice VII fine enough for the Rietveld analysis. Except for run #6, the compression axis of the NPDAC was parallel to the incident neutron beam, i.e., the incident beam penetrates one side of the anvil and the scattered neutrons mostly travel through the gasket ("through anvil geometry"). A focusing guide consisting of 1.0 m-long parabolic supermirrors with critical angles of 4.0 $Q_c$ ($Q_c$ is the critical momentum transfer for Ni) was placed in front of NPDAC to increase the neutron flux at the sample position. $^3$He gas detectors positioned in a pair of banks at $2\theta = 90°$ (coverage: $2\theta = 90 \pm 11.3°$, $\chi = 0 \pm 34.6°$)[28]. To reduce the background, radial collimators with a gauge length of 1.1 mm and of 0.5 mm for runs #1-#3 and runs #4-#7, respectively[37]. In run #6, the compression axis of the NPDAC was perpendicular to the incident beam, with incoming neutrons travelling through the gasket to reach the sample and the scattered neutrons also travelling through the gasket ("through gasket geometry"). For the "through gasket geometry", we modified NPDAC as described in Supplementary Information 3. The original data in runs #1 and #2 were already published in Komatsu et al.[14], but the detailed results and analyses are first presented here. The sample pressures were determined from the observed unit cell volume of ice using the known equation of state reported by Hemley et al.[39]. The Rietveld analyses were performed using the GSAS[31] with EXPGUI[40] packages on the diffraction patterns containing mixed phases of sample, diamond (anvils), and iron (gasket) if necessary. Detailed experimental conditions, all diffraction patterns and results of the Rietveld analyses for respective runs are presented in Supplementary Information 1.

## Data availability

The data supporting the findings of this study are found in the Supporting Information, or publicly available from the corresponding author upon request.

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

## Acknowledgements

We thank Dr. Katsutoshi Aoki, Dr. Shinji Tsuneyuki, and Dr. Jun Tsuchiya for fruitful discussion. Neutron diffraction experiments were performed through the J-PARC user programmes (Nos. 2016I0011, 2017I0011, 2019I0011, 2019B0119, 2020B0083, 2021B0047, 2021B0159, 2022A0326, 2023A0077). Preliminary test experiments of x-ray diffraction for pressure generation of NPDAC with through gasket geometry (not reported) were conducted in the PF user programme (No. 2020PF-39). This study was supported by JSPS KAKENHI (Grant Numbers:

21K18154 to K.K. and T.I., 20H01998 to A.S.-F., 20K05425 to S.M., 18H05224 to H.Kagi, K.K., and A.S.-F., 18H01936 to K.K. and S.M., 15H05829 to T.I. and K.K.), JSPS-CNRS bilateral joint research project (SAKURA grant no. 29717X to K.K. and S.K. and PRC grant no. 2191 to S.K.), and the Joint Usage/Research Center PRIUS, Ehime University, Japan. The vanadium and gadolinium metals for intensity corrections and incident neutron collimation, respectively, were machined by the laser processing machine at the CROSS laboratory. K.Y., H.Kobayashi and H.I. are recipients of MERIT, MERIT-WINGS and IGPEES programmes of the University of Tokyo for support, respectively.

## Author contributions

K.K. and S.K. conceived and designed the experiments. K.K., S.K., S.M., K.Y., T.I. and T.S. developed the nano-polycrystalline diamond anvil cells. T.H. developed the radial collimator. K.K., T.H., S.K., S.M., K.Y., H.I., H.Kobayashi, A.S.-F. and H. Kagi conducted the neutron diffraction experiments. K.K. and A.S.-F. conducted the preliminary x-ray diffraction experiments. K.K. analysed neutron diffraction patterns with contributions from T.H., S.K., S.M., K.Y., H.I. and H.Kobayashi. K.K. wrote the manuscript with contributions from all co-authors.

## Competing interests

The authors declare no competing interests.
