## [Peer Review File · Nature Communications]

Hydrogen bond symmetrisation in D2O ice observed by neutron diffractionREVIEWER COMMENTS

Reviewer #1 (Remarks to the Author):

see the attached PDF

Reviewer #2 (Remarks to the Author):

These are difficult experiments to carry out which needs to be acknowledged. Unfortunately, I have major concerns regarding the reliability and interpretation of the data.

My comments (including some minor comments) are:

- * The recently discussed bcc superionic ice is not mentioned in the introduction.
- * The discussion in the paragraph starting with "Six independent runs..." is too technical. Such details should be moved to the Supplementary Information.
- * Why are $x(D)$ and $U(D)$ strongly correlated? This is simply a statement without any evidence. Is it actually really meaningful to use isotropic displacement parameters in this study? Isn't much of this work about the fact that the hydrogen atoms are distributed along the O-O distances, ie in a highly anisotropic fashion?
- * The labels (a) and (b) seem to have shifted in Figure 2?
- * I don't think the fit in the lower part of Figure 2 is very good. Can this be improved?
- * Figure 3 shows the isotropic displacement parameters. Much work on ice VII has been done to describe the multi-site disorder. This is ignored here. Also, why does Figure 3 only go up to 40 GPa?
- * The extrapolation to 120 GPa in Figure 4(a) should not be done. From the data shown here, it is really not clear that the two O-D distances would converge. Equally, the "crossing" in Figure 4(b) includes some severe extrapolation. Combined with the problems surrounding the displacement parameters, I don't think that much can be said.
- * The conclusion from this manuscript seems to be that ice X has actually NOT been made. I don't think the description of the precursor form X' is reliable. In particular since there are these strong correlations between position and displacement parameters. There is also quite substantial variation between the different runs.
- * I also note that the diamond reflections are very strong in the data. At some pressures, the diamond and ice Bragg peaks overlap. This can introduce a lot of uncertainty.
- * In summary, I am not convinced that a clear and reliable picture emerges. Basically, I think it can be argued that ice VII is simply compressed up to quite high pressures and then the transition to ice X is not really seen.

In their ms “hydrogen bond symmetrisation in D2O ice observed by neutron diffraction”, Komatso et al. describe novel neutron diffraction experiments into the megabar range and shed new light on the elusive Ice VII-X transition.

This ms represents a tremendous technical feat in the field of neutron diffraction which was usable up to about 25 GPa beforehand. This alone would be reason enough to recommend publication in Nature Comm. The observed, admittedly sluggish, transition from a bimodal to unimodal deuterium probability distribution is not necessarily novel or ground-breaking but it certainly adds up to the bulk of literature focusing on this topic, and deserves publication as well.

The ms is well written and kept in clear language. The figures are nice and well made (even though I have some suggestions in terms of data representation) and almost all relevant recent literature has been cited. There are a couple of things I would like to point out which needs a bit of polishing but apart from that, I would wholeheartedly recommend publication.

Major points:

1.) the majority of the argument backing up the observation of the ice VII-X transition is given in the main text figure 4. In particular figure 4b and 4c are a bit unusual to me. There seems to be significant scattering in the experimental runs #1 and #2. where is this scattering coming from? In particular in light of the relatively clean looking runs #4 and #5? Run #3 is not plotted in 4b at all even though this is one of the two runs crossing over 100 GPa, that a quite remarkable omission and raises the question if the data is really so good as they seem. Runs #1 and #2 seem to scatter so much that it appears that all runs are not reproducible and only #4 and #5 seem to coincide somewhat.

Furthermore, only for #4 the ADPs could be refined, which only reached 40 GPa (about half the pressure of the to observed transition). Even though the authors claim that the extrapolation to 100 GPa is justified, it still seems a bit far fetched. A possible way around this could be to simulate several P2(x) maps for ADPs varying over say +/- 50% from the values used to produce figure 4a and show the interested reader in a supplemental figure that the optimistic extrapolation to 100 GPa is indeed justified.

2.) I think the way the data in figure 4c is represented is unfortunate. Maybe it would be better to normalize $\Delta d/d$ against their respective maximum values, so normalize all runs to 1. other wise it is very difficult to really get a lot of information out of this plot other than that the data scatters a lot. Also I would recommend colors of more contrast to each other. The figure is so packed with data points, it will be very hard to read when it gets printed in two-column mode of any journal.

minor points:

- 1) page 1, paragraph 1, line 11: the parenthesis with the quotation is weird, please cite properly, e.g. “... than in deuterium[7]. ”
- 2) page 1, paragraph 2, line 10: same issue with the awkward citation
- 3) figure 1: I think it should be written “time average” and not “time ave”, right?
- 4) Page 2, paragraph 1, line 5: typo: strongly
- 5) figure 2: please consider how the figure will look in print. I guess the text in the figure will be very hard to read. I strongly recommend polishing this figure a bit.
- 6) Figure 3: the information content of figure 3 is quite low. Maybe one could combine it with figure 2 or move it to the supplementary entirely. And again, please consider how they will appear in print.
- 7) Figure 4a: it seems like data points from runs #1, #2 and #6 as well as the ones from Guthrie et

al. are faded. I would strongly recommend, for the sake of clarity, to show all of the taken data. It doesn't matter if they do not fit so well with the $P_2(x)$ map, it would still be a compelling figure.

- 8) Figure 4b: kind of the same issue here, due to the color scheme chosen by the authors, only run #4 and #5 really catch the eye. The other runs are kind of out of focus, which seems not really honest.
- 9) Same figure: I am not so sure if the extrapolation of $2\sqrt{U}$ needs to be shown like this. In particular with the limited data set reaching only 40 GPa. A shaded bar, much like the one already in the figure, could suffice as long as the authors make it crystal clear in the caption what they have been doing.
- 10) Page 3: this is one big paragraph. Please get some structure in there. It is very hard to read otherwise.
- 11) Page 3, further down the page, I think the author of the NMR papers is spelled with an "i" rather than a "y".
- 12) page 4, first paragraph, line two: typo: "shear" instead of "shere"
- 13) page 4 paragraph 2: please also cite the computational work from Trybel et al. (10.1103/PhysRevB.102.184310), this work is related to the conclusions of the paper and should be included in the discussion

Answers to the reviewers' comments

We thank the editor for conducting the review process and the referees for their helpful comments.

We are glad to see that both reviewers acknowledge the quality of our work, and we recognise that major concerns from reviewers, mainly regarding the uncertainty of ADP and extrapolation to higher pressure than observed data, should be carefully considered. Then we decided to collect an additional dataset (Run#7) to extend the observed ADPS to higher pressures. We have addressed all the concerns and have made necessary changes to the original manuscript and supplementary information.

Reviewer #1

In their ms "hydrogen bond symmetrisation in D₂O ice observed by neutron diffraction", Komatsu et al. describe novel neutron diffraction experiments into the megabar range and shed new light on the elusive Ice VII-X transition. This ms represents a tremendous technical feat in the field of neutron diffraction which was usable up to about 25 GPa beforehand. This alone would be reason enough to recommend publication in Nature Comm. The observed, admittedly sluggish, transition from a bimodal to unimodal deuterium probability distribution is not necessarily novel or ground-breaking but it certainly adds up to the bulk of literature focusing on this topic, and deserves publication as well. The ms is well written and kept in clear language. The figures are nice and well made (even though I have some suggestions in terms of data representation) and almost all relevant recent literature has been cited. There are a couple of things I would like to point out which needs a bit of polishing but apart from that, I would wholeheartedly recommend publication.

Response:

We thank the referee for reviewing our manuscript and are pleased to hear your positive and thoughtful comments, which greatly polish our manuscript.

Major points:

1.) the majority of the argument backing up the observation of the ice VII-X transition is given in the main text figure 4. In particular figure 4b and 4c are a bit unusual to me. There seems to be significant scattering in the experimental runs #1 and #2. where is this scattering coming from? In particular in light of the relatively clean looking runs #4 and #5? Run #3 is not plotted in 4b at all even though this is one of the two runs crossing over 100 GPa, that a quite remarkable omission and raises the question if the data is really so good as they seem. Runs #1 and #2 seem to scatter so much that it appears that all runs are not reproducible and only #4 and #5 seem to coincide somewhat.

Furthermore, only for #4 the ADPs could be refined, which only reached 40 GPa (about half the pressure of the to observed transition). Even though the authors claim that the extrapolation to 100 GPa is justified, it still seems a bit far fetched. A possible way around this could be to simulate several P2(x) maps for ADPs varying over say +/- 50% from the values used to produce figure 4a and show the interested reader in a supplemental figure that the optimistic extrapolation to 100 GPa is indeed justified.

A1) The reason why runs #1, #2 are so scattered, #3 was missing, whereas #4, #5 are well reproducible is simply due to the difference in the sample preparation. Initially, in runs #1-#3, Ice VII samples were just prepared by direct compression at room temperature, but we realised that the obtained ice was coarse-grained and not suitable for Rietveld analyses. Thus, in runs #4-#7, we prepared samples through low temperature, i.e., i) D₂O water was loaded in the NPDAC and clamped nearly ambient pressure, ii) the whole NPDAC was cooled down by dry ice or liquid nitrogen, iii) compressed at low-temperature, and iv) reheated to room temperature under pressure. This procedure passes through many phase boundaries of ice phases and allows us to make a fine powder of ice VII. We reported the

result of the Rietveld analysis for run #1 in the previous manuscript, since it was still analysable. But, they are such scattered as you indicated, so that we omitted the results of #1 in the current manuscript. The description of the low-temperature procedure (as written above with underlining) is added in the Methods section. Also, we added a comment for this in the main text as follows. (last line in p4)

“Fine powder samples good enough for the Rietveld analyses were obtained in runs #4 - #7 (see details in Methods section).”

Furthermore, we conducted an additional run (#7) to extend the pressure region to observe both $x(D)$ and $U(D)$. The obtained results up to 67.5 GPa (Figs. 3 and 4) are consistent with the reported ones in the original manuscript, confirming that the main result – ice VII' - X' transition pressure would be around 80 GPa – does not change. The observed $2\sqrt{U(D)}$ in #7 may scatter between 0.2 and 0.3 as shown in the grey shaded area in the modified Fig. 4b, say $\pm 25\%$ of the centred value (~ 0.25). Considering that the observed $d(D...O)$ is also scattered within ± 0.1 Å (Fig. 4b), the transition pressure (cross points of $2\sqrt{U(D)}$ and $d(D...O)$ lines) would vary from 70 to 90 GPa. We believe the results from the additional run would be sufficient to determine the transition pressure. Also, we roughly estimated the confidence limit for the $U(D)$ would be $\pm 25\%$ from χ^2 map with varying $x(D)$ and $U(D)$ as shown below, which also helps to convince our results and address your comments on several additional $P_2(x)$ maps with variation of ADPs. This consideration with the above figures is described in Supplementary Information “2. Reliability and correlations of structural parameters”.

Supplementary Figure 9. The χ^2 values after the Rietveld analyses using GSAS⁵ with varying atomic coordinate, $x(D)$, and, isotropic ADP, $U(D)$, for ice VII (a) at 8.22 GPa (run #4, exp #78831) and (b) at 7.67 GPa (run #5, exp #79064). The parameter range for $x(D)$ is from 0.4 to 0.439 with 0.01 step, and that for $U(D)$ is from 0.005 to 0.054 with 0.001 step, so that 2000 χ^2 values in total are plotted as density maps.

2.) *I think the way the data in figure 4c is represented is unfortunate. Maybe it would be better to normalize $\Delta d/d$ against their respective maximum values, so normalize all runs to 1. otherwise it is very difficult to really get a lot of information out of this plot other than that the data scatters a lot. Also I would recommend colors of more contrast to each other. The figure is so packed with data points, it will be very hard to read when it gets printed in two-column mode of any journal.*

A2) We appreciate your helpful suggestion. We modified Fig. 4c (now it is Fig. 3c) for visibility, for example, used simple black and red colours for high contrast, and also omitted error bars for simplicity. On the other hand, $\Delta d/d$ data were not scaled but shown as they are, because the scaling was not that helpful for visibility and $\Delta d/d$ themselves have physical meaning representing the microstrain. What we would like to stress in this figure is not the individual points behaviour of respective runs, but the overall trends in which the peak widths of 110 decreased suddenly at ~80 GPa, while the 111 peak widths had no such features at ~80 GPa in contrast. We hope the modified figure is clear enough to show these features.

minor points:

1) *page 1, paragraph 1, line 11: the parenthesis with the quotation is weird, please cite properly, e.g. "... than in deuterium[7]. "*

We modified it as suggested.

2) *page 1, paragraph 2, line 10: same issue with the awkward citation*

We modified "as ice VII^{11,12}".

3) *figure 1: I think it should be written "time average" and not "time ave", right?*

We modified it as suggested.

4) *Page 2, paragraph 1, line 5: typo: strongly*

We modified it as indicated.

5) *figure 2: please consider how the figure will look in print. I guess the text in the figure will be very hard to read. I strongly recommend polishing this figure a bit.*

We appreciate your helpful suggestion. We modified Fig. 2 – avoid to using subscripts in legends and use larger fonts, plus delete left-hand figures showing anvil shapes and variable parameters with the crystal structure of ice VII since they were a bit too technical. The anvil shapes are moved to Supplementary Figures.

6) *Figure 3: the information content of figure 3 is quite low. Maybe one could combine it with figure 2 or move it to the supplementary entirely. And again, please consider how they will appear in print.*

We agree with your suggestion, Figure 3 is moved to Supplementary (reindexed as Supplementary Figure 8), and modified for visibility.

7) *Figure 4a: it seems like data points from runs #1, #2 and #6 as well as the ones from Guthrie et al. are faded. I would strongly recommend, for the sake of clarity, to show all of the taken data. It doesn't matter if they do not fit so well with the $P2(x)$ map, it would still be a compelling figure.*

We carefully considered this matter and decided not to show data points from runs #1 – #3, because ice VII samples in these runs were not goos powder (random orientation with sufficiently small particles) but coarse crystals. The ice samples in runs #4 – #7 are prepared through low temperature shown in the response to your question 1). This procedure was written in Supplementary Information in the original manuscript, but more clearly described in the method section in the modified version. Then, the results of Rietveld refinements for #1 - #3 are also deleted since they are not reliable.

8) *Figure 4b: kind of the same issue here, due to the color scheme chosen by the authors, only run #4 and #5 really catch the eye. The other runs are kind of out of focus, which seems not really honest.*

We modified the colour scheme as suggested, and also included the data points from runs #4 – #7 for $d(D...D)$ and #4 and #7 for $2\sqrt{U}$.

9) *Same figure: I am not so sure if the extrapolation of $2\sqrt{U}$ needs to be shown like this. In particular with the limited data set reaching only 40 GPa. A shaded bar, much like the one already in the figure, could suffice as long as the authors make it crystal clear in the caption what they have been doing.*

We extended the data points for $U(D)$ up to ~70 GPa from additional run #7, which gives more confidence in the extrapolation. At least, we believe that it can be

10) *Page 3: this is one big paragraph. Please get some structure in there. It is very hard to read other wise.*

We divided the paragraph into several paragraphs as suggested.

11) *Page 3, further down the page, I think the author of the NMR papers is spelled with an “i” rather than a “y”.*

12) *page 4, first paragraph, line two: typo: “shear” instead of “shere”*

We thank you for your indication of typos which were corrected.

13) *page 4 paragraph 2: please also cite the computational work from Trybel et al. (10.1103/PhysRevB.102.184310), this work is related to the conclusions of the paper and should be included in the discussion*

We thank your suggestions, and we found that Trybel et al. mentioned the important issue regarding the non-existence of sharp phase transition. We discussed this issue in the paragraph as shown below.

“More recent DFT calculations for H₂O ice by Trybel et al.³⁶ also showed a gradual change from the double- to a single-well potential, symmetrisation starts above 90 GPa accompanying the significant drop of proton jump frequencies, and fully symmetric single well potential is expected above 130 GPa. These “transition” pressures are roughly consistent with the crossing pressure point of $d(D-D)$ and $2\sqrt{U(D)}$ at around 80 GPa, and the merging point of the extrapolated $d(O-D)$ and $d(D...O)$ at around 120 GPa, respectively. Trybel et al. noted that their calculation did not show any sharp phase transition in the pressure range from 2 GPa to 200 GPa, at least there is no structural

phase transition of first or second order. (skip several lines) As indicated by Trybel et al.³⁶, it is worth pursuing the issue that how sharp structural or thermodynamic parameters change can be observed in the step-wise changes from VII to X including intermediate VII' and X'. The current understanding would be that the VII'-X' transition may be second order or higher since the elastic constants show discontinuity²⁸, whereas VII-VII' and X'-X transitions could be third order or higher since they only show the pressure dependences of bulk moduli¹⁰."

Reviewer #2

These are difficult experiments to carry out which needs to be acknowledged. Unfortunately, I have major concerns regarding the reliability and interpretation of the data.

Response:

We appreciate reviewer #2 for reviewing our manuscript and recognising the difficulty of our experiments. We carefully read your comments and accepted your criticism that our previous manuscript would not have enough reliability and needs improvement for the interpretation of our data. Therefore, we conducted an additional experiment run (run #7) to observe the ADPs of deuterium ($U(D)$) at as high pressure as possible and managed to obtain the structural parameters including $U(D)$ up to 67.5 GPa, well achieving the potential VII'-X' transition as discussed below. We hope the additional experiments and the modification of the manuscript are satisfactory for publication.

My comments (including some minor comments) are:

1) The recently discussed bcc superionic ice is not mentioned in the introduction.

We thank your suggestion and insert the following short introduction about the super-ionic ices in the introduction. "the phase diagram of ice is dominated by body-centred cubic (bcc) ices with hydrogen disordered ice VII, ordered ice VIII and hydrogen-bond (H-bond) symmetrised ice X, and recently discussed superionic ices having body-centred cubic and face-centred cubic structures^{2,3}"

2) The discussion in the paragraph starting with "Six independent runs..." is too technical. Such details should be moved to the Supplementary Information.

We moved the technical description to the method section as suggested.

3) Why are $x(D)$ and $U(D)$ strongly correlated? This is simply a statement without any evidence. Is it actually really meaningful to use isotropic displacement parameters in this study? Isn't much of this work about the fact that the hydrogen atoms are distributed along the O-O distances, ie in a highly anisotropic fashion?

We appreciate your thoughtful comments, which help to polish our understanding. In our previous manuscript, we just mentioned the tendency that $x(D)$ and $U(D)$ to be correlated. We carefully checked the correlation and found that $x(D)$ and $U(D)$ have a positive correlation, but are not as strong as we thought before. We realised that the main reason for unstable refinements is not from the correlation but from the small contribution from $U(D)$ to the diffraction intensity and due to the smaller sample volume in runs #5 and #6. We discussed this issue in Supplementary Information "2. Reliability and correlations of structural parameters", and also in the main text as follows.

"This was done since the sample volumes in runs #5 and #6 are much smaller than those in #4 and #7 (see Supplementary Table 1), and $x(D)$ and the small deviation of $U(D)$ does not contribute sufficient to the diffraction intensities to allow stable refinements."

In terms of the (an)isotropy of ADPs of deuterium, note that there are two sites of deuterium, and the duplication of two site distributions flexibly describes the anisotropy of deuterium distributed along the O...O direction, even if each site distribution is approximated as isotropic. Of course, there may be deviation from the isotropic manner to some extent, our results should be a kind of standard since all previous neutron diffraction studies for ice VII obey

single-site deuterium model with isotropic ADPs including Guthrie et al. who achieve the highest pressure up to 60 GPa.

Supplementary Figure 9. The χ^2 values after the Rietveld analyses using GSAS⁵ with varying atomic coordinate, $x(D)$, and, isotropic ADP, $U(D)$, for ice VII (a) at 8.22 GPa (run #4, exp #78831) and (b) at 7.67 GPa (run #5, exp #79064). The parameter range for $x(D)$ is from 0.4 to 0.439 with 0.01 step, and that for $U(D)$ is from 0.005 to 0.054 with 0.001 step, so that 2000 χ^2 values in total are plotted as density maps.

4) The labels (a) and (b) seem to have shifted in Figure 2?

We thank your indication. We modified Figure 2 and it was moved to Supplementary Figure 8 according to the suggestion from Reviewer #1.

5) I don't think the fit in the lower part of Figure 2 is very good. Can this be improved?

Considering the error of ADPs (say as large as $\pm 25\%$, judging from the 3σ from the refinements), we would say it would be difficult to improve the fit. We would avoid stating this in the main text since it is unclear, but there may be some anomalies at around 15-30 GPa in $U(O)$, as found in many structural and physical properties in previous studies. The fitting results for $U(O)$ do not affect our main results because we only use $U(D)$, $x(D)$, and $x(O)$ to discuss where the hydrogen-bond symmetrisation occurs, please see Fig. 3b in the main text.

6) Figure 3 shows the isotropic displacement parameters. Much work on ice VII has been done to describe the multi-site disorder. This is ignored here. Also, why does Figure 3 only go up to 40 GPa?

We acknowledge your indication that the multi-site disorder was not mentioned in the previous manuscript. We recently conducted single crystal and powder neutron diffraction to clarify the multi-site disorder (Yamashita et al., 10.1073/pnas.2208717119), and revealed that the single-site model adopted in this study and in most of all previous studies reproduces well by the Rietveld analysis even for the high-quality diffraction pattern. To elucidate the small deviation of oxygen from the 2a site at (0.25, 0.25, 0.25), high-quality data with high-Q diffraction peaks and sophisticated data analysis like the maximum entropy method are necessary. We added a comment about why we adopted the single-site model in the main text (P5).

“The oxygen atom was fixed at a single site 2a, (1/4,1/4,1/4). Although multi-site disorder models for the oxygen were proposed^{15,16}, the displacement of oxygen from the 2a site was too small to elucidate from the limited resolution data. Thus, the multi-site disorder of oxygen was ignored in this study, and in fact, it is known from our recent neutron diffraction study¹⁷ that the single site model well reproduced the observed diffraction pattern.”

7) The extrapolation to 120 GPa in Figure 4(a) should not be done. From the data shown here, it is really not clear that the two O-D distances would converge. Equally, the "crossing" in Figure 4(b) includes some severe extrapolation. Combined with the problems surrounding the displacement parameters, I don't think that much can be said.

We agreed with your comment that our previous extrapolation was too far from the observed one. Then, we collected additional data as higher pressures and with quality as high as possible. In the additional run (#7), we managed to obtain the $U(D)$ up to 67.5 GPa and the trend of $U(D)$ agrees well with the previous extrapolation as shown in the modified Fig. 3a (used to be 4a). Plus, we also plot the $d(O-D)$ and $d(D...O)$ from the recent DFT calculation by Tsuchiya et al., showing excellent agreement at least up to 100 GPa. We believe our additional data to 67.5 GPa are now sufficiently close to the VII'-X' transition pressure at around 80 GPa. Regardless of how $U(D)$ deviate from the extrapolation, the transition pressure would be within 80 ± 10 GPa, which should be a strong constraint to the possible transition scenario.

8) The conclusion from this manuscript seems to be that ice X has actually NOT been made. I don't think the description of the precursor form X' is reliable. In particular since there are these strong correlations between position and displacement parameters. There is also quite substantial variation between the different runs.

As described in the previous responses, we conducted an additional run and confirmed our conclusion. We also added some discussion concerning the relation to the recent DFT study by Trybel et al. as shown below, their interpretation shows excellent consistency to our data.

“More recent DFT calculations for H₂O ice by Trybel et al.³⁵ also showed a gradual change from the double- to a single-well potential, symmetrisation starts above 90 GPa accompanying the significant drop of proton jump frequencies, and a fully symmetric single well potential is expected above 130 GPa. These “transition” pressures are consistent with the crossing pressure point of $d(D-D)$ and $2\sqrt{U(D)}$ at around 80 GPa, and the merging point of the extrapolated $d(O-D)$ and $d(D...O)$ at around 120 GPa, respectively.”

9) I also note that the diamond reflections are very strong in the data. At some pressures, the diamond and ice Bragg peaks overlap. This can introduce a lot of uncertainty.

Your indication was partly right, the ice 110 peak overlapped with the diamond 111 peak from 35 to 60 GPa. However, we conducted at least four individual runs including completely different cell set-ups (#6, through gasket geometry)

to other runs (through anvil geometry) and all the results show the same trend (Fig. 3).

10) In summary, I am not convinced that a clear and reliable picture emerges. Basically, I think it can be argued that ice VII is simply compressed up to quite high pressures and then the transition to ice X is not really seen.

We did not insist that our observation shows the transition to ice X (in a narrow definition, meaning that the atomic coordinate of the deuterium atom comes exactly at the centre of the hydrogen bond and deuterium distributes perpendicular (oblate distribution) to the O...O direction), but we first experimentally confirmed the deuterium distribution above 100 GPa, and found that the distribution changed from bimodal (VII') to unimodal (X') at around 80 GPa. In other words, we propose throughout this manuscript to distinguish ice X' at 80–120 GPa (having unimodal deuterium distributions) and ice X above 120 GPa (having $d(\text{O}\dots\text{O}) = 2d(\text{O}-\text{D})$ in the space- and time-averaged structure models from diffraction data) even though both have symmetric hydrogen bonds. Note here that the unimodal distribution can be realised even when two sites are not completely merged (bottom in Fig. R1), since the actual distribution is the combination of two peaks and this state is exactly called ice X'.

Figure R1. Simulations of atomic distribution function, $P_2(x)$ (eq. 2), with varying $d(\text{D}\dots\text{D})$ and $U(\text{D})$ parameters. Actual atomic distribution shown as red lines are combinations of contributions from two crystallographically equivalent sites shown as dotted lines.

Our observations should be a strong constraint to possible transition scenarios. For example, the possibility that the symmetrisation occurs at 30 GPa (suggested by x-ray diffraction) can be ruled out. We believe that our result are a major progress in the understanding of the transition of ice VII to symmetric ice X which should be made known to the ice community.

REVIEWER COMMENTS

Reviewer #1 (Remarks to the Author):

All comments were addressed accordingly. I strongly back-up publication of the manuscript. Good work.

Reviewer #2 (Remarks to the Author):

Dear Editor,

I am generally happy with the revisions made by the authors. As a sanity check, it would be worth exploring if refining the highest pressure data with one hydrogen position and anisotropic Uiso would give meaningful fits. It would be interesting to see if the Uiso ellipsoid elongates along the O-O direction (as would be expected). I would then be happy for the article to be published.

Response to the reviewer 1's comment

Comment from reviewer 1

All comments were addressed accordingly. I strongly back-up publication of the manuscript. Good work.

Response

We sincerely appreciate your strong support. Your comments greatly improve the clarity and visibility of our manuscript.

Response to the reviewer 2's comment

Comment from reviewer 2

I am generally happy with the revisions made by the authors. As a sanity check, it would be worth exploring if refining the highest pressure data with one hydrogen position and anisotropic Uiso would give meaningful fits. It would be interesting to see if the Uiso ellipsoid elongates along the O-O direction (as would be expected). I would then be happy for the article to be published.

Response

We would like to thank the reviewer for her/his suggestion. We have analyzed the data taken at 103 GPa (the highest pressure point with long-exposure time) based on the model that the deuterium position is fixed to be the (pseudo-)centre with varying anisotropic ADPs. If the deuterium atom is fixed to exactly centre ($x = 0.5$), the anisotropic ADPs are constrained by the site symmetry so that the atomic distribution would be fully spherical. Therefore, we fixed the deuterium position to be 0.4999, slightly off from the exact centre, with half occupancy. The result gives $U_{\parallel}(\text{D}) = 0.021(2) \text{ \AA}^2$, $U_{\perp}(\text{D}) = 0.017(2) \text{ \AA}^2$, showing almost sphere or slightly elongate distribution along the O...O direction, as you expected. This result is well consistent with our original refinement based on the two D-sites model with isotropic ADP, however, the two models are hard to distinguish from each other since they give almost identical consistencies with the diffraction patterns.

We added a few sentences on this issue in the main text as follows (red coloured, added sentences).

P8

We note that a spherical atomic distribution of two D-sites at distances $-d/2$ and $d/2$ is indistinguishable by diffraction measurements of a single H atom at the centre, but with an ellipsoidal atomic distribution with its long axis along the O...O direction. **We conducted such refinements with varying anisotropic ADP for the data taken at 103 GPa in run #5, yielding $U_{\parallel}(\text{D}) = 0.021(2) \text{ \AA}^2$, $U_{\perp}(\text{D}) = 0.017(2) \text{ \AA}^2$, showing almost sphere or slightly elongate distribution along the O...O direction. The χ^2 of the refinement based on the anisotropic ADPs is almost identical to that based on the two-site model with isotropic ADP (see more details in Supplementary Information 6).**

Also, the detailed methodology for the analysis is described in Supplementary Information 6.

Anisotropic ADPs, U_{ij} , are constrained by the site symmetry to have two components, one parallel to the O-D direction, $U_{\parallel}(\text{D})$, and one perpendicular, $U_{\perp}(\text{D})$, in ice VII structure³, such that the following conditions should be obeyed.

$$\begin{aligned} U_{11} = U_{22} = U_{33} &= \frac{1}{3}U_{\parallel} + \frac{2}{3}U_{\perp} \\ U_{12} = U_{13} = U_{23} &= \frac{1}{3}U_{\parallel} - \frac{1}{3}U_{\perp} \end{aligned} \quad (\text{S6.3})$$

Thus, the two components are given by anisotropic ADPs,

$$U_{\parallel} = U_{ii} + 2U_{ij}, U_{\perp} = U_{ii} - U_{ij} \quad (\text{S6.4})$$

where $i \neq j$.

We have conducted the Rietveld analysis with varying anisotropic ADPs for the data taken at 103 GPa. In the refinement, we fixed the deuterium position to be $x = 0.4999$ with half occupancy instead of the completely centred position ($x = 0.5$) with full occupancy, since the anisotropic ADPs will be constrained by the site symmetry to be exactly spheric when $x = 0.5$. The χ^2 of the refinement based on the (pseudo) one-site model with the anisotropic ADPs is 7.104, whereas that based on the two-site model with the isotropic ADP is 7.095, so the two models give almost identical consistency to the obtained diffraction pattern taken at 103 GPa.

We hope these modifications satisfy as a response to your suggestion.

Sincerely yours,

REVIEWERS' COMMENTS

Reviewer #2 (Remarks to the Author):

The authors have successfully conducted the final 'sanity' test I asked them to perform. I recommend the article is published.